# Discrete Tree Flows via Tree-Structured Permutations

**Mai Elkady** [*1]  **Jim Lim** [*2]  **David I. Inouye** [2]

## Abstract

While normalizing flows for continuous data have been extensively researched, flows for discrete data have only recently been explored. These prior models, however, suffer from limitations that are distinct from those of continuous flows. Most notably, discrete flow-based models cannot be straightforwardly optimized with conventional deep learning methods because gradients of discrete functions are undefined or zero, and backpropagation can be computationally burdensome compared to alternative discrete algorithms such as decision tree algorithms. Previous works approximate pseudo-gradients of the discrete functions but do not solve the problem on a fundamental level. Our approach seeks to reduce computational burden and remove the need for pseudo-gradients by developing a discrete flow based on decision trees—building upon the success of efficient tree-based methods for classification and regression for discrete data. We first define a tree-structured permutation (TSP) that compactly encodes a permutation of discrete data where the inverse is easy to compute; thus, we can efficiently compute the density value and sample new data. We then propose a decision tree algorithm to learn TSPs that estimates the tree structure and simple permutations at each node via a novel criteria. We empirically demonstrate the feasibility of our method on multiple datasets.

## 1. Introduction

Discrete data is abundant in numerous applications and domains, from DNA sequences and medical records to text data and many forms of tabular data. Learning a density model for discrete data is a fundamental task useful for understanding or leveraging such data. However, discrete data can be hard to model especially when the number of categories and the dimension of the data is large. Prior methods include probabilistic graphical models (Inouye et al., 2014; 2017; Wainwright & Jordan, 2008), VAEs (Kingma & Welling, 2019), or autoregressive models (Germain et al., 2015). However, these works either lack exact likelihood computation or require expensive sampling procedures.

As an alternative, discrete flows provide exact likelihood computation and efficient sampling. The two key components of discrete flows are a base distribution (similar to continuous flows) and a permutation of the discrete configuration values (a discrete version of invertible functions) (Tran et al., 2019). In its full generality, this requires optimizing over the set of all permutations, whose size is doubly exponential in terms of the feature dimension. Thus, parameterizing and optimizing over a set of possible permutations is critical for practical algorithms and generalizability. Tran et al. (2019) propose to parameterize these permutations by neural networks and optimize them using standard backpropagation. Tran et al. (2019) estimate pseudo-gradients of discrete functions using the straight-through gradient estimator (Bengio et al., 2013) along with a Gumbel-softmax distribution for backpropagation, which is an approximation of the forward pass equation (i.e., one-hot of argmax function). van den Berg et al. (2020) later show that the architecture of the coupling layers is significantly more important than the gradient bias issue. However, none of these discrete flow methods address the discrete nature of the problem on a fundamental level and may be computationally expensive compared to alternative discrete-oriented algorithms such as those based on decision trees—which have seen wide success in classification and regression for discrete data (e.g., XGBoost (Chen & Guestrin, 2016)). Thus, we seek to answer the following research question: **Can we design a more computationally efficient discrete flow algorithm using decision trees that handles discrete data on a fundamental level?**

To answer this, we propose a novel tree-structured permutation (TSP) model that can compactly represent permutations and a novel decision tree algorithm that optimizes

---

[*]Equal contribution  [1]Department of Computer Science, Purdue University, Indiana, USA [2]School of Electrical and Computer Engineering, Purdue, Indiana, USA. Correspondence to: Mai Elkady <melkady@purdue.edu>, Jim Lim <lim316@purdue.edu>.

Third workshop on *Invertible Neural Networks, Normalizing Flows, and Explicit Likelihood Models* (ICML 2021). Copyright 2021 by the author(s).

over the space of these permutations. Moreover, for more powerful permutations, we can iteratively build up a sequence of TSPs to form a deep permutation. We also propose a novel decision tree algorithm for learning TSPs that includes a novel splitting criteria and an efficient algorithm for determining the best permutation at each tree node.

**Background: Discrete Flows**  The Discrete flow change of variables formula is given by: $P_x(x) = Q_z(f(x))$, where $Q_z$ is some known base distribution that is usually a simple distribution (e.g. an independent categorical distribution), and $f$ is an invertible function. Importantly, in the discrete case, the invertible function can only be a permutation of the possible discrete configurations of $x$ so there is no change in volume, hence no need to incorporate the Jacobian determinant term as done in continuous flow. In its full generality, discrete flows would require optimizing over the set of all possible permutations, whose size is doubly exponential in terms of the dimension—which emphasizes the computational intractability of the general problem. Thus, two key ingredients for practical discrete flows are a simple and compact parameterization of permutations and an algorithm to optimize over these permutations. Tran et al. (2019) propose two different architectures (autoregressive flows and bipartite flows) to parameterize the discrete flow permutations. Hoogeboom et al. (2019) and van den Berg et al. (2020) propose integer discrete flows for compression of image-based data (rather than categorical tabular data as in this paper).

## 2. Tree-Structured Permutations

Our goal is to define a set of permutations that is both *computationally tractable* (in terms of evaluating the permutation and its inverse and optimizing over the set) and *generalizable* (i.e., the class of permutations is more likely to generalize well to new test data). To achieve this goal, we introduce tree-structured permutations (TSP) as new model for discrete flows that utilizes trees for compactly parameterizing a set a of permutations.

**Notation**  We will denote a discrete dataset as $X \in \mathcal{Z}^{n \times d}$ where $n$ is the number of samples, $d$ is the number of dimensions, and $\mathcal{Z}$ is a set of discrete values, and where $k$ is the maximum number of possible discrete values per feature (i.e., the number of categories). We will denote tree nodes by $\mathcal{N}$. The split information at each node $\mathcal{N}$ will be encoded by a feature index $s \in \{0, 1..., d-1\}$ and a split value $v$ such that data with the $s$-th feature equal to $v$ will go to the left node and all other data will go to the right node. We define the node *domain*, denoted by $\mathcal{D}(\mathcal{N})$, to be the set of all discrete configurations that could reach this node when traversing the decision tree. We will denote permutations by $\pi(\cdot) \in \Pi$, where $\Pi$ will denote a set of

permutations. Given that the number of discrete configurations of $d$ features with $k$ possible discrete values is $k^d$, the number of all possible permutations is $(k^d)!$, which is $O(\exp(k^d))$, i.e., doubly exponential in $d$. We will denote the parameters of the permutation by $\theta_\pi$, where the exact parametrization may depend on context.

### 2.1. Definition of TSP

A *tree-structured permutation* is a binary decision tree where each node $\mathcal{N}$ contains both a permutation $\pi$ and the usual split information (i.e., a split feature $s$ and split value $v$). To evaluate a TSP, an input vector traverses the tree from the root to a leaf node based on the split information and applies node permutations as soon as it reaches the node (i.e., the node permutation will be applied before determining the split). More formally, we can define the evaluation of a TSP recursively as the evaluation of a node $f_\mathcal{N}$ where the initial $\mathcal{N}$ is the root node:

$$f_\mathcal{N}(x) = \begin{cases} x & \text{if } \mathcal{N} \text{ is a leaf node} \\ f_{\text{left}(\mathcal{N})}(\pi_\mathcal{N}(x)) & \text{if } [\pi_\mathcal{N}(x)]_j = v \\ f_{\text{right}(\mathcal{N})}(\pi_\mathcal{N}(x)) & \text{otherwise} \end{cases} \quad (1)$$

where $f_\mathcal{N}$ is the evaluation of a node, $f_{\text{left}(\mathcal{N})}$ denotes the evaluation of the left child node (and similarly for the right node), $\pi_\mathcal{N}$ is the permutation associated with the node, and $[\pi_\mathcal{N}(x)]_j = v$ denotes the condition that the $j$-th feature after the permutation has value $v$. We illustrate the idea of the forward traversal with an example in Figure 1. Note that even though each node's permutation is invertible, it is unclear whether the entire forward evaluation of a TSP is invertible (i.e., if the path and sequence of permutations implied by the tree structure can be inverted)—which is critical for their use in discrete flows. Thus, we explore the conditions for the invertibility of TSPs.

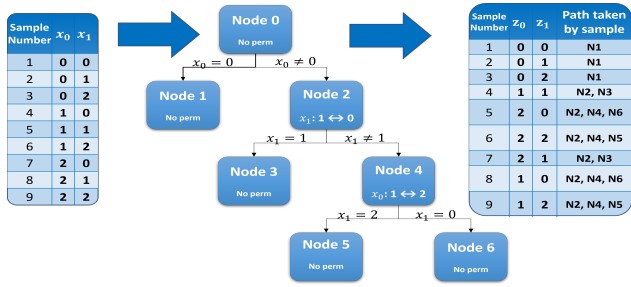

*Figure 1.* This example of evaluating a TSP on input data (left) to compute the output (right) illustrates that our tree can compactly represent a permutation. At each node, the input data will be permuted by the node's associated permutation and then pass onto the left or right nodes depending on the split information.

### 2.2. Invertibility Constraint and Calculation for TSPs

To ensure our TSPs are invertible (and thus applicable to discrete flows), we prove that a simple and intuitive con-

straint on the node permutations is sufficient for invertibility (in the appendix, we derive a necessary and sufficient condition for invertibility but it is challenging to check or enforce).

**Theorem 1** (TSP Invertibility Constraint). *A TSP is invertible if all node permutations $\pi_\mathcal{N}$ do not permute configurations that are outside of the node's domain, i.e., $\pi_\mathcal{N}(x) = x, \forall x \notin \mathcal{D}(\mathcal{N})$.*

We will denote the set of permutations that satisfy this constraint for a node $\mathcal{N}$ as $\Pi(\mathcal{N})$. The proof of the if direction is constructive and relies on the following lemma.

**Lemma 2.** *If this invertibility constraint is satisfied, the TSP tree traversal path for any input can be recovered from the output.*

Leveraging Lemma 2, the inverse can be computed by traversing the tree in the forward direction (i.e., from root to leaves) without applying the node permutations but keeping track of the node path. Once we reach a leaf, we apply the permutations along the path in reverse order to compute the inverse. We illustrate this inverse computation using the example output point $z = [2, 0]$ from Figure 1. Starting at the root, we see that because $z_0 \neq 0$, $z$ will move right to node 2. Then, at node 2, the node permutation will be ignored (for the inverse) and the split condition again will be checked on $z$ such that $z$ will then move to node 4 because $z_1 \neq 1$, etc. Finally, the inverses of permutations along the path will be applied in reverse order.

### 2.3. Naïve TSP: Restricting to the class of independent permutations

For computational tractability and generalizability, we will further restrict the possible permutations at each node to be feature-wise independent (i.e., the permutation for one feature does not depend on values of other features) and, for $k > 2$, value-pair swaps (i.e., permute by swapping one pair of categorical values). We will denote these restrictions as $\pi \in \Pi_{\text{IndValPair}}$, where the size of the permutation space is $O(k^{2d})$. Because of independence of the permutation and the base distribution, however, the computational cost can be reduced to $O(dk^2)$ (more details in the appendix). Additionally, while we could permute all $d$ features independently at each tree node, we choose to only permute the single best feature at each node to avoid overfitting the dataset, denoted by $\Pi_{\text{TSP}}(\mathcal{N}, s)$, where $s$ is the feature index of the single best feature to permute. An example of this naïve TSP is shown in Figure 1. To increase model expressiveness despite these restrictions, we propose to compose a sequence of naïve TSPs into a *deep* TSP permutation that we call Discrete Tree Flows (DTF).

## 3. Learning TSP Discrete Flows

We will present a greedy approach for building up the tree similar to other decision tree methods where the goal is to minimize the negative log likelihood (NLL) over naïve TSP permutations and independent base distributions $Q_z$ given our dataset $\mathcal{X}$:

$$\min_{Q_z \in \mathcal{Q}_{\text{Ind}}, \pi \in \Pi_{\text{TSP}}} \mathcal{L}(\pi, Q_z) = \min_{Q_z \in \mathcal{Q}_{\text{Ind}}, \pi \in \Pi_{\text{TSP}}} \sum_{i=1}^n -\log Q_z(\pi(x_i)),$$

where $\mathcal{L}(\pi, Q_z)$ denotes the negative log-likelihood, $\mathcal{Q}_{\text{Ind}}$ is the set of independent distributions over categorical data, and $\Pi_{\text{TSP}}$ is the set of naïve TSP permutations. Assuming that the base distribution is independent enables efficient methods for determining the best permutations and the best features using only feature-wise category counts at each node. Additionally, an independent base distribution enables fast sampling, unlike autoregressive base distributions or other complex base distributions. Our proposed learning algorithm can be decomposed into node-wise subproblems where two main steps are required. First, we need to determine the best permutation for each node. Second, as with all decision tree algorithms, we need to determine the best node to split among all current leaf nodes (i.e., a splitting criteria). We describe our objectives for both of these steps in the next section.

### 3.1. Permutation Criteria for Naïve TSP construction

Our method of naïve TSP construction revolves around building a tree that minimizes the negative log-likelihood (NLL) of our training data. We select the best single-feature node permutation $\pi$ that minimizes the change in the NLL:

$$\min_{s, \pi \in \Pi_{\text{TSP}}(\mathcal{N}, s)} \Delta_\mathcal{L}(\mathcal{N}, s, \pi) \tag{2}$$

$$= \min_{s, \pi \in \Pi_{\text{TSP}}(\mathcal{N}, s)} \mathcal{L}(\pi, Q_z') - \mathcal{L}(\text{id}, Q_z) \tag{3}$$

where $Q_z'$ is a refitted base distribution after applying permutation $\pi$, id denotes that no permutation is applied, and $\Pi_{\text{TSP}}(\mathcal{N}, s)$ denotes the class of independent permutations that act only on the domain of the node (so that the TSP will be invertible) and restricted to the $s$-th feature. Note that because $Q_z'$ is an independent base distribution, the optimal $Q_z'$ can be computed efficiently only using the empirical (and smoothed) frequencies of each feature for the training data at the current node. Given that each feature is independent (both in the permutation and the base distributions), we can greatly simplify the computation of $\Delta_\mathcal{L}$ (details in appendix).

### 3.2. Splitting Criteria for Naïve TSP construction

To determine the best node split, we consider using a splitting criteria that maximizes the difference between the fac-

torized distribution on the left and the factorized distribution on the right of the proposed split. Intuitively, if the distributions on the right and left are different, then a permutation will be able to align them better (and thus help reduce NLL). Let us denote $Q_{\text{left}}(s, v)$ and $Q_{\text{right}}(s, v)$ to be the best *independent* distributions on the left and right of a proposed split parametrized by $s$ and $v$. We want to maximize the divergence between these two distributions, where we will use generalized Jensen-Shannon Divergence (JSD). The full derivation is given in the appendix, and the splitting criteria is:

$$s^*, v^* = \underset{s,v}{\arg\min}\; w_1 H(Q_{\text{left}}^{(s,v)}) + w_2 H(Q_{\text{right}}^{(s,v)}), \quad (4)$$

where $w_1$ and $w_2$ are the relative empirical probability of data going left versus right. As is standard in decision tree algorithms, we also take the minimum over all current leaf nodes $\mathcal{N}$ to select the next best leaf node to split.

### 3.3. TSP construction: Training the model

Training our model entails greedily constructing the TSP based on our training data by determining the node permutation, split feature $s$ and split value $v$ at each node $\mathcal{N}$. The tree construction takes in a maximum height of the tree given by the user ($M$), and constructs the TSP nodes node by node, until the maximum height is reached or until no further splits can be made (which can be due to reaching a very small number of samples at each node, or running out of viable split features or values). At a low-level, the algorithm mainly requires keeping track of feature-wise counts at each node and a global count vector to optimize each criteria. The computational complexity for the algorithm is $O(2^{2M} n d^2 k^2)$, where $M$ is the max depth of the tree, but the quadratic $d^2$ term can be reduced to $d$ using a Monte Carlo approximation (details can be found in the appendix).

## 4. Experiments

We compared our model against the two models introduced in (Tran et al., 2019) and implemented in (Bricken, 2021). All the models including ours use an independent base distribution. We use the following abbreviations in our comparisons, AF for autoregressive flow, BF for bipartite flow. The BF model implemented in (Bricken, 2021), however, had some issues and so we implemented some modifications to it (that we mention in more details in the appendix).

For the first set of experiments (labelled exp=1 through 5), we generated synthetic data $X \in \mathcal{Z}^{n \times d}$ with $k$ categories and split the data into training and validation datasets using cross validation with 5 folds. For $d = 2$ and $k = 2$, we chose to generate data that follows a known probability distribution. Specifically, for exp = 1, $d = 2, k = 2$, we use the probabilities $\Pr([0,0]) = 1/3, \Pr([0,1]) =$

$1/6, \Pr([1,0]) = 1/6, \Pr([1,1]) = 1/3$, and for exp = 2, $d = 2, k = 2$, we use the probabilities $\Pr([0,0]) = 1/8, \Pr([0,1]) = 3/8, \Pr([1,0]) = 3/8, \Pr([1,1]) = 1/8$. For the other combinations of $d$ and $k$, we simulated the probability distribution over all possible configurations as a Dirichlet distribution (with $\alpha = 1$) and we simulate $n = 10000$ data points. We also ran an experiment for a real discrete dataset[1], that has $n = 8124$ datapoints which is denoted "exp=6". For all exps we report the mean and standard deviation of NLL across the 5 test folds. For the DTF, the results are for a flow of 1 TSP where the maximum depth $M$ of the TSP is 2 for experiments 1 and 2, and $M = 4$ for the rest. For the AF flows, we used a single flow, that utilizes 64 hidden units (in the MADE layers). We ran this model for a total of 1500 epochs, sampling 1024 data points in each epoch. Corresponding to the configuration in (Tran et al., 2019), 4 flow layers were used for the BF model. As can be observed from Table 1, our method gives comparable results to that of AF and BF models.

*Table 1.* NLL averaged across the 5 test folds.

| Experiments | DTF | | AF | | BF | |
| details | NLL | Std | NLL | Std | NLL | Std |
|---|---|---|---|---|---|---|
| exp = 1, d = 2, k = 2 | 1.3333 | 0.0036 | 1.3338 | 0.0035 | 1.3664 | 0.0249 |
| exp = 2, d = 2, k = 2 | 1.2521 | 0.0117 | 1.2527 | 0.0119 | 1.3291 | 0.0719 |
| exp = 3, d = 5, k = 5 | 8.0358 | 0.0033 | 8.0301 | 0.0067 | 8.003 | 0.0058 |
| exp = 4, d = 10, k = 5 | 16.0832 | 0.0021 | 16.2613 | 0.0219 | 16.1122 | 0.0061 |
| exp = 5, d = 5, k = 10 | 11.5008 | 0.0024 | 11.5739 | 0.0145 | 11.5412 | 0.0068 |
| exp = 6, d = 22, k = 12 | 16.6367 | 1.6347 | 24.9785 | 2.9568 | 27.1999 | 2.5411 |

For exp = 6, our model took 5.4372 sec for training (with an Std of 2.463 sec), the AF model took 39.5838 sec (with an Std of 0.2519 sec), and the BF model took 143.8929 sec (with an Std of 2.2670 sec). For exp 1-5, timing results are available in the appendix. Thus, our approach seems to be computationally less expensive than prior methods as we had hoped though further experimentation is likely needed.

## 5. Discussion

We presented a novel framework for discrete normalizing flows that relies on tree-structured permutations (TSPs), which we define and develop. Our model isn't without limitations though, as our implementation relies on a greedy algorithm and thus may end up not achieving a global optimum when learning deep naïve TSPs. This is a problem that is similar in nature to problems faced by most decision tree algorithms. Moreover, handling very high dimensional data may be computationally challenging. However, previous decision tree algorithms were able to overcome these obstacles, and we expect that similar techniques could be used in our method. Ultimately, we hope that our paper lays the groundwork for developing practical and effective discrete flows using decision tree algorithms.

---

[1] https://archive.ics.uci.edu/ml/datasets/Mushroom

## 6. Acknowledgement

The authors acknowledge support from the Army Research Lab through Contract number W911NF-2020-221.

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

# A. Invertibility Proofs

## A.1. Proof of Lemma 2 in Section 2.2

**Lemma 2.** *If the invertibility constraint is satisfied, the TSP tree traversal path for any input can be recovered from the output.*

*Proof.* First, note that because $\pi$ is a permutation (i.e., one-to-one mapping) and $\pi_{\mathcal{N}}(x) = x, \forall x \notin \mathcal{D}(\mathcal{N})$, then $\forall \boldsymbol{x} \in \mathcal{D}(\mathcal{N}), \pi_{\mathcal{N}}(\boldsymbol{x}) \in \mathcal{D}(\mathcal{N})$, i.e., all node permutations $\pi_{\mathcal{N}}$ do not permute configurations from in the domain to outside the domain.

Without loss of generality, we consider trees where all leaf nodes are at the max depth of $M$.

We will denote the tree traversal path of an input $\boldsymbol{x}$ up to tree level $i$ to be $\mathcal{P}_i(\boldsymbol{x}) \triangleq (\mathcal{N}_{L_0(\boldsymbol{x})}, \mathcal{N}_{L_1(\boldsymbol{x})}, \cdots, \mathcal{N}_{L_i(\boldsymbol{x})})$, where $\mathcal{N}_{L_j(\boldsymbol{x})}$ is the tree node that $\boldsymbol{x}$ reaches at $j$-th level of the tree (i.e., $L_j(\boldsymbol{x})$ gives the index of the node for the particular input $\boldsymbol{x}$ at the $j$-th level), $\mathcal{N}_{L(0)}$ is the root node. Let $\boldsymbol{y} \triangleq \pi_{\text{TSP}}(\boldsymbol{x})$ denote the output of our TSP forward evaluation.

For all $\boldsymbol{x} \in \mathcal{Z}^d$, let $\boldsymbol{y} \triangleq \pi_{\text{TSP}}(\boldsymbol{x}) \equiv \pi_{\mathcal{N}_{L_M(\boldsymbol{x})}} \circ \cdots \circ \pi_{\mathcal{N}_{L_1(\boldsymbol{x})}} \circ \pi_{\mathcal{N}_{L_0(\boldsymbol{x})}}(\boldsymbol{x})$ denote the output of our TSP forward evaluation. We want to prove that $\mathcal{P}_M(\boldsymbol{x})$ can be recovered from $\boldsymbol{y}$ and $\pi_{\text{TSP}}$. Let $\mathcal{P}'_i(\boldsymbol{y}) \triangleq (\mathcal{N}_{L'_0(\boldsymbol{y})}, \mathcal{N}_{L'_1(\boldsymbol{y})}, \cdots, \mathcal{N}_{L'_i(\boldsymbol{y})})$ where we traverse the decision tree of $\pi_{\text{TSP}}$ *without* applying node permutations and where $L'_j(\boldsymbol{y})$ is the index of that $\boldsymbol{y}$ reaches at the $j$-th layer. If we can prove $\mathcal{P}'_M(\boldsymbol{y}) = \mathcal{P}_M(\boldsymbol{x})$ or equivalently $\forall j, L'_j(\boldsymbol{y}) = L_j(\boldsymbol{x})$, then we are done.

**Inductive hypothesis** For all $i \in \{0, 1, \cdots, M\}$, $\mathcal{P}'_i(\boldsymbol{y}) = \mathcal{P}_i(\boldsymbol{x})$.

**Base case ($i = 0$)** Since both $\boldsymbol{y}$ and $\boldsymbol{x}$ start at the root node, then the path up to $i = 0$ is the same.

**Induction step** We need to prove that if $\mathcal{P}'_i(\boldsymbol{y}) = \mathcal{P}_i(\boldsymbol{x})$, then $\mathcal{P}'_{i+1}(\boldsymbol{y}) = \mathcal{P}_{i+1}(\boldsymbol{x})$. From assumption of inductive hypothesis, we know that $\mathcal{N}_{L'_i(\boldsymbol{y})} = \mathcal{N}_{L_i(\boldsymbol{x})}$ (i.e., $\boldsymbol{x}$ and the corresponding $\boldsymbol{y}$ are the same node for level $i$). Let $\boldsymbol{x}^{(i)} = \pi_{\mathcal{N}_{L_i(\boldsymbol{x})}} \circ \cdots \circ \pi_{\mathcal{N}_{L_1(\boldsymbol{x})}} \circ \pi_{\mathcal{N}_{L_0(\boldsymbol{x})}}(\boldsymbol{x})$ where $\boldsymbol{x}^{(M)} \equiv \boldsymbol{y}$. Note that only the value $v$ of the split feature $s$ for both $\boldsymbol{x}^{(i)}$ and $\boldsymbol{y}$ is relevant for determining whether to go left or right.

We prove that the chosen nodes are the same using contradiction. Suppose $x_s^{(i)} = v$ such that $\boldsymbol{x}$ goes left and suppose $y_s \neq v$ such that $\boldsymbol{y}$ would go right.

We know that the $s$-th part of the domain of the left child has only $v$ in it, i.e., $\mathcal{D}_s(\mathcal{N}_{\text{left}}) = \{v\}$. Also, we know that domains of children are always smaller disjoint subsets of

the parent, i.e., $\mathcal{D}_s(\mathcal{N}_{L_N(\boldsymbol{x})}) \subseteq \mathcal{D}_s(\mathcal{N}_{\text{left}})$ for all $N > i$.

Thus, $x_s^{(N)} = v$ for all $N > i$ because the invertibility constraint ensures that we cannot permute a value inside the domain to a value outside the domain. However, this is a contradiction to our assumption that $x_s^{(M)} \equiv y_s \neq v$. Therefore, if $\boldsymbol{x}$ goes left, then $\boldsymbol{y}$ will also go left.

In a similar way, now suppose $x_s^{(i)} \neq v$ such that $\boldsymbol{x}$ goes right and suppose $y_s = v$ such that $\boldsymbol{y}$ would go left. The $s$-th part of the domain of the right child has $\mathcal{D}_s(\mathcal{N}_{\text{right}}) = \{a : a \neq v, a \in \mathcal{D}_s(\mathcal{N})\}$. Again, $\mathcal{D}_s(\mathcal{N}_{L_N(\boldsymbol{x})}) \subseteq \mathcal{D}_s(\mathcal{N}_{\text{right}})$ for all $N > i$ because every child is a subset of the parent domain.

Therefore, $\boldsymbol{x}_s^{(N)} \in \mathcal{D}_s(\mathcal{N}_{\text{right}})$ for all $N > i$, and thus in particular $x_s^{(M)} \in \mathcal{D}_s(\mathcal{N}_{\text{right}})$, where $x_s^{(M)} \equiv y_s$ by definition. However, this contradicts our assumption that $y_s = v$ (i.e., goes left) because $v \notin \mathcal{D}_s(\mathcal{N}_{\text{right}})$.

Hence, if $x$ goes right, $y$ will also go right.

Combining these two we get that $L'_{i+1}(\boldsymbol{y}) = L_{i+1}(\boldsymbol{x})$ (i.e., they will both go left or both go right), and thus we can recover the path for $i + 1$ by adding the child node to the path for $i$, i.e., $\mathcal{P}'_{i+1}(\boldsymbol{y}) = \mathcal{P}_{i+1}(\boldsymbol{x})$. This proves our inductive step and concludes the proof of the lemma. $\square$

## A.2. Proof of Theorem 1 in Section 2.2

**Theorem 1** (TSP Invertibility Constraint (sufficient only))**.** *A TSP is invertible if all node permutations $\pi_{\mathcal{N}}$ do not permute configurations that are outside of the node's domain, i.e., $\pi_{\mathcal{N}}(x) = x, \forall x \notin \mathcal{D}(\mathcal{N})$.*

*Proof.* Lemma 2 (proven above) states that the TSP traversal path for any input can be recovered from the output. Thus, the output path is identical to the input path $\mathcal{P}'_i(\boldsymbol{y}) = \mathcal{P}_i(\boldsymbol{x})$. Therefore, $\boldsymbol{x} = \pi_{\text{TSP}}^{-1}(\boldsymbol{y}) \equiv \pi_{\mathcal{N}_{L'_0(\boldsymbol{y})}}^{-1} \circ \cdots \circ \pi_{\mathcal{N}_{L'_{M-1}(\boldsymbol{y})}}^{-1} \circ \pi_{\mathcal{N}_{L'_M(\boldsymbol{y})}}^{-1}(\boldsymbol{y})$ because each node permutation is itself invertible by the definition of a permutation. This can be seen as traversing the tree from the corresponding leaf node to the root node and applying the inverse node permutations along the path. $\square$

## A.3. Necessary and Sufficient Condition for Invertibility with Proof

We now present a corrected condition (i.e., disjoint ranges of leaf nodes) that is both necessary and sufficient for invertibility. Note that this new theorem can be easily used to prove Theorem 1 as a corollary because the original invertibility constraint set is a subset of the disjoint range of leaf nodes constraint. However, the proof here does not provide an efficient algorithm for determining the leaf node

while the proof of Lemma 2 does provide an efficient algorithm (i.e., merely traverse the tree as described in the lemma proof to determine the path).

First, we define the range of a node to simplify the definition of the theorem and proof.

**Definition 1** (Range of a Node). *We define the* range *of a node, denoted $\mathcal{R}(\mathcal{N})$, as the image of $\mathcal{D}(\mathcal{N})$ under $\pi_{\mathcal{N}}$, i.e., $\mathcal{R}(\mathcal{N}) \triangleq \{\pi_{\mathcal{N}}(\boldsymbol{x}) : \boldsymbol{x} \in \mathcal{D}(\mathcal{N})\}$.*

Now we present our theorem with a corrected condition that is both necessary and sufficient.

**Theorem 3** (TSP Necessary and Sufficient Invertibility Constraint). *A TSP is invertible if and only if the range of each leaf node is disjoint from all other leaf nodes, i.e., $\mathcal{R}(\mathcal{N}) \cap \mathcal{R}(\mathcal{N}') = \emptyset, \forall \mathcal{N}, \mathcal{N}' \in \mathcal{T}_{\text{leaf}}$ such that $\mathcal{N} \neq \mathcal{N}'$, where $\mathcal{T}_{\text{leaf}}$ are the set of leaves in the TSP tree.*

*Proof.* We use a constructive proof for the if direction (sufficiency). Because each of the permutations themselves are invertible, the primary challenge is showing that we can find the right path through the tree (as there could be multiple paths without constraints). If the disjoint range condition is satisfied, then each possible output can be mapped to one of the leaves, i.e., $\mathcal{N}_{L_M(\boldsymbol{y})}$ is the leaf node such that $\boldsymbol{y} \in \mathcal{R}(\mathcal{N})$. Given the leaf node, there is only one possible path through the decision tree back to the root. Thus, the inverse can be computed by traversing from the leaf node to the root node and applying the inverse of each node's permutation.

To prove the only-if direction (necessity), we will use a proof by contradiction. Suppose a TSP is invertible but the disjoint range condition is not satisfied, then $\mathcal{R}(\mathcal{N}) \cap \mathcal{R}(\mathcal{N}') \neq \emptyset$. Therefore, there exists an output $\boldsymbol{y}$ that is in the range of two leaf nodes, i.e., $\exists \boldsymbol{y}$ such that $\boldsymbol{y} \in \mathcal{R}(\mathcal{N})$ and $\boldsymbol{y} \in \mathcal{R}(\mathcal{N}')$ where $\mathcal{N} \neq \mathcal{N}'$. Yet, each input traverses the TSP tree in a deterministic way and thus each unique input will always arrive at the same leaf node. Therefore, there must exist two distinct inputs $\boldsymbol{x} \neq \boldsymbol{x}'$ such that $\pi_{\text{TSP}}(\boldsymbol{x}) = \pi_{\text{TSP}}(\boldsymbol{x}') = \boldsymbol{y}$. This means that two distinct inputs map to the same output (i.e., not one-to-one mapping) and violates invertibility. However, this contradicts our assumption that the TSP is invertible. $\square$

## B. A discussion of classes of permutations

Since the number of discrete configurations of $d$ features with $k$ possible discrete values is $k^d$, the number of all possible permutations is $(k^d)!$, which is $O(\exp(k^d))$, i.e., doubly exponential in $d$. We chose to make some restrictions to our general permutations class as described below.

**Independent feature-wise permutations** We choose to restrict to the natural and computationally tractable class

of independent feature-wise permutations, denoted $\Pi_{\text{Ind}}$, which allows each feature to be permuted independently of the other features (i.e., the permutation of one feature cannot depend on the permutations of other features). This class significantly reduces the number of permutations compared to all possible permutations, i.e., $|\Pi_{\text{Ind}}| = (k!)^d \approx \exp(kd) \ll \exp(k^d) \approx (k^d)! = |\Pi|$. For example, for binary data ($k = 3$) and three features ($d = 5$), we can have 3! permutations for each of the five features, leading to a total of $(3!)^5 = 7776$ joint permutations—which is considerably smaller than the total number of permutations $(3^5)! = 243! \approx 10^{474}$.

This makes the independent class of permutations significantly more computationally tractable and more likely to generalize to new data points. Additionally, if we assume that the base distribution $Q_z$ is independent, it is possible to separate the optimization problem into subproblems for each feature that can be solved independently. Thus, each feature subproblem only needs to evaluate $k!$ permutations, and thus the computational complexity for our problem can be reduced to $O(d \cdot k!)$.

**Permutations that swap a single pair of values for $k > 2$** While restricting to independent permutations $\Pi_{\text{Ind}}(\mathcal{N})$ can significantly reduce our computational complexity to $O(dk!)$, the computationally complexity would still grow exponentially in terms of $k$, the number of features. Thus, we further restrict the class of node permutations to be the class of permutations that swaps a single pair of the $k$ possible values (or categories) while holding all other feature values constant. We will denote this class by $\Pi_{\text{IndValPair}}$ and this reduces number of possible permutations per feature from $k!$ to $\binom{k}{2} + 1$ which is $O(k^2)$ (where the +1 is to include the identity permutation). Thus, with this restriction and our assumption about $Q_z$ being independent, we can reduce the computational complexity to $O(dk^2)$, which is computationally in all parameters including $k$.

As an example, if $k = 3$, then the possible permutations in this class are $(012, 102, 210, 021)$ which correspond to the original permutation, swapping 0 and 1, swapping 0 and 2, and swapping 1 and 2.

As mentioned in the main section, in our Naïve TSP implementation, we restrict the node permutations to $\pi \in \Pi_{\text{IndValPair}}$. Additionally, while we could permute all $d$ features independently at each tree node, we choose to only permute the single best feature at each node to avoid overfitting the dataset; we leave exploration of permuting multiple features to future work.

We denote this single-best permutation class as $\Pi_{\text{TSP}}$, where $s$ is the feature index of the single best feature to permute. An example of this naïve TSP is shown in Figure 1.

## C. Derivation for Calculating the difference in NLL

We can greatly simplify the computation of $\Delta_{\mathcal{L}}$ as follows:

$$\Delta_{\mathcal{L}}(\mathcal{N}, s, \pi_{\mathcal{N}}) \triangleq \mathcal{L}(\pi_{\mathcal{N}}, Q'_z) - \mathcal{L}(\text{id}, Q_z) \tag{5}$$

$$= \frac{1}{n} \sum_{i=1}^{n} \sum_{j=1}^{d} -\log Q'_{z_j}([\pi_{\mathcal{N}}(\boldsymbol{x}_i)]_j) - \\ \frac{1}{n} \sum_{i=1}^{n} \sum_{j=1}^{d} -\log Q_{z_j}(\boldsymbol{x}_{ij}) \tag{6}$$

$$= \frac{1}{n} \sum_{i=1}^{n} \sum_{j \neq s} -\log Q'_{z_j}(\boldsymbol{x}_{ij}) - \log Q'_{z_s}([\pi_{\mathcal{N}}(\boldsymbol{x}_{is})]_s) \\ -\frac{1}{n} \sum_{i=1}^{n} \sum_{j \neq s} -\log Q_{z_j}(\boldsymbol{x}_{ij}) - \log Q_{z_s}(\boldsymbol{x}_{is}) \tag{7}$$

$$= \frac{1}{n} \sum_{i=1}^{n} \sum_{j \neq s} -\log Q_{z_j}(\boldsymbol{x}_{ij}) - \log Q'_{z_s}([\pi_{\mathcal{N}}(\boldsymbol{x}_{is})]_s) \\ -\frac{1}{n} \sum_{i=1}^{n} \sum_{j \neq s} -\log Q_{z_j}(\boldsymbol{x}_{ij}) - \log Q_{z_s}(\boldsymbol{x}_{is}) \tag{8}$$

$$= \frac{1}{n} \sum_{i=1}^{n} -\log Q'_{z_s}([\pi_{\mathcal{N}}(\boldsymbol{x}_{is})]_s) + \log Q_{z_s}(\boldsymbol{x}_{is}) \tag{9}$$

Where $id$ denotes that no permutations applied are applied (the identity), and $Q'_z$ is the independent base distribuition estimate after $\theta_\pi$ is applied.

Thus, while constructing the tree, for every node $\mathcal{N}$ we seek to minimize the following term.

$$s^*, \pi^* = \underset{s \in d, \pi \in \Pi_{\text{TSP}}}{\arg\min} \Delta_{\mathcal{L}}(\mathcal{N}, s, \pi_{\mathcal{N}}), \tag{10}$$

Where $s^*$ is the best permutation feature and $\pi^*$ is the best permutation to apply.

To simplify Eqn.9, we notice that it can be rewritten if we applied the following properties, for a particular feature (s):

$$\sum_{i=1}^{n} \log Q_{z_s}(\boldsymbol{x}_{is}) \\ = \sum_{c=0}^{k-1} \log Pr(\boldsymbol{x}_{is} = c)^{n_c} \tag{11} \\ = \sum_{c=0}^{k-1} n_c \log Pr(\boldsymbol{x}_{is} = c)$$

Where $n_c$ is the number of samples that has the category c in the $s$ dimension index.

This property can be used in Eqn.9 to re-write it as:

$$\frac{1}{n} \sum_{i=1}^{n} -\log Q'_{z_s}([\pi(\boldsymbol{x}_{is})]_s) + \log Q_{z_s}(\boldsymbol{x}_{is}) \tag{12}$$

$$= \frac{1}{n} \sum_{c=0}^{k-1} -n'_c \log Pr'(\boldsymbol{x}_{is} = c) + n_c \log Pr(\boldsymbol{x}_{is} = c) \tag{13}$$

$$= -\frac{1}{n} \sum_{c=0}^{k-1} n'_c \log\left(\frac{n'_c + 1}{n + k}\right) - n_c \log\left(\frac{n_c + 1}{n + k}\right) \tag{14}$$

$$= -\frac{1}{n} \sum_{c=0}^{k-1} n'_c \log(n'_c + 1) - n_c \log(n_c + 1) \\ - (n'_c - n_c) \log(n + k) \tag{15}$$

Where $n'_c$ is the number of samples with category c in the $s$ dimension after applying the permutation $\pi_s$

Furthermore, since for naïve TSPs, we have restricted our permutations to $\Pi_{\text{TSP}}$ which not only allow us to be computationally more efficient but also allow us to further simplify the calculation of $\Delta$. Let's say that the permutation $\pi_s$ flips values $k = v_1$ (that appears on the left child) with $k = v_2$ (that appears on the right child) then the number of counts only in the categories that are equal to $v_1$ and $v_2$ will change before and after applying the permutation, while the counts of the remaining categories will remain constant. This means that Eqn(15) can be re-written as

$$-\frac{1}{n} [n'_{v_1} \log(n'_{v_1} + 1) - n_{v_1} \log(n_{v_1} + 1) - \\ (n'_{v_1} - n_{v_1}) \log(n + k) + n'_{v_2} \log(n'_{v_2} + 1) \\ -n_{v_2} \log(n_{v_2} + 1) - (n'_{v_2} - n_{v_2}) \log(n + k)] \tag{16}$$

where $n'_{v_1}$ and $n'_{v_2}$ are number of counts of categories $v_1$ and $v_2$ respectively after applying the permutation $\pi_s$ and $n_{v_1}$ and $n_{v_2}$ are number of counts of categories $v_1$ and $v_2$ respectively before applying the permutation $\pi_s$. For all the other categories where $k \neq v_1$ and $k \neq v_2$, $n'_c$ will be equal to $n_c$, resulting in the summation in Eqn(15) being zero for those categories.

## D. Derivation for calculating the splitting criteria

As mentioned in the main section, to determine the best node split, we consider using a splitting criteria that maximizes the difference between the factorized distribution on the left and the factorized distribution on the right of the proposed split. Intuitively, if the distributions on the

right and left are different, then a permutation will be able to align them better (and thus help reduce NLL). Let us denote $Q_{\text{left}}(s, v)$ and $Q_{\text{right}}(s, v)$ to be the best *independent* distributions on the left and right of a proposed split parametrized by $s$ and $v$. We want to maximize the divergence between these two distributions, where we will use generalized Jensen-Shannon Divergence (JSD). The generalized JSD can be written either as a weighted sum of KL divergences or a difference in entropy terms:

$$\text{JSD}(P, Q; \boldsymbol{w}) \triangleq w_1 \text{KL}(P, w_1 P + w_2 Q)$$
$$+ w_2 \text{KL}(Q, w_1 P + w_2 Q) \tag{17}$$

$$\equiv H(w_1 P + w_2 Q) - w_1 H(P) - w_2 H(Q). \tag{18}$$

This second version can be seen as the entropy of the mixture minus the mixture of the entropies. Given this definition, we can now define our splitting problem as:

$$\max_{s,v} \text{JSD}(Q_{\text{left}}^{(s,v)}, Q_{\text{right}}^{(s,v)}; \boldsymbol{w}^{(s,v)})$$
$$= H(Q_{parent}) + \max_{s,v} -w_1 H(Q_{\text{left}}^{(s,v)}) - w_2 H(Q_{\text{right}}^{(s,v)}) \tag{19}$$

where $Q_{parent} = w_1 Q_{\text{left}}^{(s,v)} + w_2 Q_{\text{right}}^{(s,v)}$, $Q_{\text{left}}^{(s,v)}(\boldsymbol{x}) = \prod_{j=1}^{d} Q_{\text{left}}^{(s,v)}(x_j)$, $Q_{\text{left}}(x_s = v)^{(s,v)} = \frac{n_{s,v}^{\text{left}}+1}{\sum_v n_{s,v}^{\text{left}}+k}$ (i.e., smoothed frequencies), and $n_{s,v}^{\text{left}}$ are the number of samples where the $s$-th feature equals the value $v$ on the left side— and similarly for the right side—, and where $\boldsymbol{w}^{(s,v)} = [\Pr(x_s = v), \Pr(x_s! = v)]$ is the probability vector encoding the probability that samples go to the left or the right side of the split. Notice that the entropy of the parent node is not required to optimize the JSD.

## E. Discussion of Algorithmic complexity

The algorithmic complexity for our algorithm will depend on the number of leaves at any point during the tree construction which at its worse would be $2^M$ where $M$ is the maximum depth of the tree. The computational complexity of determining the best permutation for a node is $O(dk^2)$, which is due to being able to solve each feature independently and considering only a single swap of feature values. Thus the computational complexity of finding the node permutations for the whole tree is $O(2^M dk^2)$. The complexity of finding the best split at a node is $O(nd^2 k)$ which is due to the entropy calculation that must be computed over all features for every possible split value. However, we can choose to make a Monte Carlo approximation to the entropy calculation by only sampling a relatively small constant number of dimensions (e.g., ten or twenty) and take the average entropy over these sampled dimensions rather

than over all dimensions. Thus, we would get an unbiased approximation to the entropy—similar in spirit to SGD. This approximation allows for the complexity of finding the best split to be $O(ndk2^M)$ and since this must be done for all current leafs a the complexity is $O(ndk2^{2M})$, which makes the overall complexity of constructing a naïve TSP construction $O(2^M dk^2 + 2^{2M} ndk^2)$. Once a Naïve TSP is constructed, passing a data in the forward direction is just a matter of the traversing the tree from root to leaf while applying the permutations stored in the tree node, which is dependant in complexity on the depth of the tree, and calculating the inverse of the data utilizes the same idea with the exception of needing additional memory to keep track of the permutations that we encounter, so both operations are $O(M)$ in time complexity.

## F. Modification of the Bipartite flow code

The BF model implemented in (Bricken, 2021), had some bugs in the code (e.g., running the bipartite model for data dimensions $d > 2$ would yield a runtime error).Thus, modifications were made to fix the bipartite model code. The model's initial embedding flow layers (commonly used for NLP sequence data) were replaced by a single hidden layer network with an activation function. A ReLU activation function was incorporated into the hidden layer to add nonlinearity. The size of the hidden layer was proportional to the feature size times half the dimension ($\frac{1}{2} * k * d$), since only half of the dimension would be mapped after the split. Each flow layer does a transformation on one of the splits, therefore, at least a paired flow layers (even number of flow layers) is required for a balanced dimensional transformation.

## G. Timing Results for exp = 1 through 5

*Table 2.* Training time in seconds averaged across the 5 test folds.

| DTF | | AF | | BF | |
|---|---|---|---|---|---|
| Time | Std | Time | Std | Time | Std |
| 0.0655 | 0.0318 | 6.161 | 0.0315 | 13.8026 | 0.1057 |
| 0.0715 | 0.0335 | 6.1373 | 0.0304 | 13.9232 | 0.0543 |
| 0.7733 | 0.3645 | 11.3008 | 0.3558 | 23.4296 | 0.1263 |
| 2.5133 | 1.1839 | 19.2331 | 0.0591 | 35.1818 | 0.3065 |
| 1.6346 | 0.7683 | 16.048 | 0.3881 | 29.3238 | 0.1261 |