# OpenReview forum: "Discrete Tree Flows via Tree-Structured Permutations"
_ICML.cc/2021/Workshop/INNF — INNF+ 2021 poster_

### Official Review · Reviewer_n3tC · 2021-06-10

**Rating:** Borderline Accept
**Confidence:** 3

**Summary:**

The paper proposes both a new discrete normalizing flow as well as a training technique for it. The model is based on tree-structure permutations and is made invertible by restricting the node evaluation functions. Experiments are performed on synthetic datasets and the Mushroom dataset from the UCI MLR, and results are compared with two common baseline models.

**Justification For Rating:**

The paper introduces an interesting new flow model building their work on decision trees. This connection was then used to come up with training techniques for the flow. The paper is in general very well-written.

The paper seems to (slightly) improve over baseline architectures. In the future, more elaborate testing on more challenging datasets should be done though.

The paper accurately describes the limitations of their work, particularly about scaling to (very) high dimensions.

---

### Official Review · Reviewer_nsgg · 2021-06-10

**Rating:** Borderline Reject
**Confidence:** 3

**Summary:**

The authors propose an (greedy) approach to construct Discrete Normalizing Flows based on decision trees, called Discrete Tree Flows (DTF). Compute such Discrete Flows, in its fully generality, is computationally intractable (scales double-exponentially on the dimensions). To overcome such a computational burden, a tree-structured permutation (TSP) model and a computational algorithm are proposed.

The main ideas are: (i) Minimize the negative log likelihood (NLL) over small subset (naıve TSP permutations) of the set of all permutations rather than the full permutations set itself.  (ii) learn TSPs that estimates the tree structure and simple permutations at each node.



**Justification For Rating:**

I am very happy to be assigned to review this paper. I was not familiar with the literature on Discrete Normalizing Flows and could learn about it by reading the paper and some of the references. Therefore, all the comments/questions below are from a non-expert reviewer who just got acquaintance with the problem. I am also asking some questions below to make sure I correctly understood it.

I am not entirely convinced that the current version of the manuscript (as it is) provides a groundwork for developing practical and effective discrete flows using decision tree algorithms — as claimed by the authors, although it undoubtedly presents many good ideas that deserves to be developed further.  The authors are aware that the method proposed has still limitations.

However, I also agree that create a method in Discreate Normalizing Flows that is both computationally efficient and has good generalisation properties is a highly non-trivial problem. The field is new and every good idea is highly appreciate for the development of Discrete Normalizing Flows.

Small comments:

1. The presentation could be surely improved.  One example from the beginning: the author could introduce/motivate better the change of variables formula. I really like the presentation/notation on the discrete change of variables formula in Tan et al. 2019 or van den Berg et al. 2020.

2. It is very unfortunate the proof of Theorem 1 is not available for the review. At least for me, a really beginner in the area, it would be super nice to have didactic examples/graphics (or at least the proof) to get a better feeling at a first reading.

3. The algorithm proposed reduces considerably the computational complexity of the problem. The authors could give some theoretical insight why the model is reasonable or, at least, make more experiments in toy-models/academic examples demonstrating some of its power/limitations. The results shown in Table 1 are important, but it is super hard to grasp the problem only from these.

Questions:
(i) How close the minimum in line 117-118 (the only equation without a numbering) are from the global optimum? Is it possible to evaluate it numerically at least in idealistic/low-dimensional examples?

(ii) Why the independent class of permutations is more likely to generalize to new data points? Clearly, the manuscript explains the computational complexity advantages.

(iii) Could the authors explain the pro/coins between the proposed approach and Probabilistic Graphical Models (e.g. Belief propagation algorithm)? Also, is this related to Wekenel and Louppe’s approach (2020) for Normalizing Flows?

(iv) Does the naıve TSP permutations class can be interpret as an extra assumption on the data that each feature has to be permuted independently of the other features?

---

### Official Review · Reviewer_ZNvb · 2021-06-11

**Rating:** Accept
**Confidence:** 2

**Summary:**

This work presents a novel algorithm for discrete flow-based models based on decision trees.
Specifically, it is first argued that discrete normalizing flow transformations can be formulated through a set of permutations.
Based on this, a Tree-Structured Permutation (TSP) is proposed.
TSPs are defined as binary trees where each node undergoes a permutation before splitting.
The invertibility of this structure is investigated, and a practical greedy algorithm for maximizing the log-likelihood using such a structure is presented.

**Justification For Rating:**

### Strengths and Weaknesses

+ The main strength of the proposed approach is its novelty given that not so many works investigate discrete flow-based models on a fundamental level.
As far as this reviewer knows, most of the current approaches around discrete normalizing flows are somehow a discretization of existing flows for continuous distributions.
This paper, in contrast, takes a different route and tries to propose a model given the discrete nature of the distribution modeling.

+ The proposed method is investigated thoroughly.
First, the invertibility of the proposed transformation is established (though the formal proofs for the theoretical claims are missing).
Then, a practical greedy algorithm for training such flow-based models is presented.
Finally, it is shown through few experiments that the proposed discrete flow can result in a significant reduction of training time in contrast to well-known baselines (discrete autoregressive and bipartite flows [1]).

+ For an audience with flow-based modeling knowledge, the paper may be hard to follow.
The main reason is that the paper revolves around decision trees which are not particularly used in normalizing flows.
However, in-depth knowledge of decision trees are somehow taken for granted to understand the contents.

### Additional Feedback and Suggestions

1. To make the paper more accessible for a wider audience, it would be nice to have a figure that shows the definitions in the Notation paragraph of Section 2.
For instance, the definition of split information, permutation, and node domain is better understood via schematic form.

2. The exact definition of the permutation $\theta_\pi$ seems to be missing.
Figure 1 does not help with this also, and it adds more confusion.
It would be better to separately define $\theta_\pi$ and the notation used in Figure 1 (it took this reviewer a while to understand the meaning of $[(0, 1 \leftrightarrow 0)]$).

3. In Eq. (1), should the value of $v$ depend to the node $\mathcal{N}$ (something like $v_\mathcal{N}$)?
Furthermore, does $j$ in this equation denote to the same thing as $s$?
If so, please consider using a unified notation throughout the paper.
Also, it would be nice to use $\boldsymbol{x}$ instead of $x$ as done in the appendix.

4. In Eq. (3), the definition of $\mathrm{id}$ is missing. Please add a sentence stating that this denotes the idenitity permutation.

5. Minor typos: line 048 (an fundamental -> a fundamental), line 092 (delete the extra a), start of line 137 (node 2 -> node 1).

### Questions

+ To what extent the proposed approach can be generalized to data other than tabular, e.g., image and text?
+ Table 2 shows that as the data dimension increases, so does the standard deviation of the training time in the proposed approach.
Can you elaborate on that? What can be the main reason?

[1] Tran, Dustin, et al. "Discrete flows: Invertible generative models of discrete data." _NeurIPS_, 2019.

---

### Decision · Program_Chairs · 2021-06-14

Accept (poster)